# Multiplexed and high-throughput neuronal fluorescence imaging with diffusible probes

Syuan-Ming Guo[1,6], Remi Veneziano[1,6], Simon Gordonov[1,6], Li Li[2,3,6], Eric Danielson[1], Karen Perez de Arce[2,3], Demian Park[4], Anthony B. Kulesa[1,3], Eike-Christian Wamhoff[1], Paul C. Blainey [1,3], Edward S. Boyden [1,4,5], Jeffrey R. Cottrell[2,3]* & Mark Bathe [1,3]*

Synapses contain hundreds of distinct proteins whose heterogeneous expression levels are determinants of synaptic plasticity and signal transmission relevant to a range of diseases. Here, we use diffusible nucleic acid imaging probes to profile neuronal synapses using multiplexed confocal and super-resolution microscopy. Confocal imaging is performed using high-affinity locked nucleic acid imaging probes that stably yet reversibly bind to oligonucleotides conjugated to antibodies and peptides. Super-resolution PAINT imaging of the same targets is performed using low-affinity DNA imaging probes to resolve nanometer-scale synaptic protein organization across nine distinct protein targets. Our approach enables the quantitative analysis of thousands of synapses in neuronal culture to identify putative synaptic sub-types and co-localization patterns from one dozen proteins. Application to characterize synaptic reorganization following neuronal activity blockade reveals coordinated upregulation of the post-synaptic proteins PSD-95, SHANK3 and Homer-1b/c, as well as increased correlation between synaptic markers in the active and synaptic vesicle zones.

[1] Department of Biological Engineering, MIT, Cambridge, MA, USA. [2] Stanley Center for Psychiatric Research, Broad Institute of MIT and Harvard, Cambridge, MA, USA. [3] Broad Institute of MIT and Harvard, Cambridge, MA, USA. [4] Media Lab, MIT, Cambridge, MA, USA. [5] McGovern Institute for Brain Research, Department of Brain and Cognitive Sciences, MIT, Cambridge, MA, USA. [6] These authors contributed equally: Syuan-Ming Guo, Remi Veneziano, Simon Gordonov, Li Li. *email: cottrell@broadinstitute.org; mark.bathe@mit.edu

Neuronal synapses are the fundamental sites of electro-chemical signal transmission in the brain and the primary cellular loci of plasticity that underlie learning and memory. Synapses are composed of hundreds of proteins whose expression levels, structural organization, and turnover govern diverse aspects of brain development and neuronal circuit function[1,2]. Because numerous synaptic protein genes have been implicated in psychiatric and neurological diseases[3–5] and synaptic protein expression levels are known to vary widely across organisms, brain regions, and neuronal cell sub-types, characterizing synaptic protein composition in situ is of major importance to both basic and translational neuroscience. While fluorescence imaging offers the opportunity to characterize the heterogeneity in synaptic protein expression levels and localizations within intact neuronal samples[6], it has been limited by its inability to visualize more than four protein species in any given neuronal sample using conventional imaging approaches.

Multiplexed protein imaging strategies that are used to overcome the spectral limit of conventional fluorescence microscopy typically involve multiple rounds of antibody staining and imaging achieved either by antibody elution[7,8] or fluorophore inactivation using photo- and/or chemical bleaching[9–11]. Array Tomography (AT) applies volumetric imaging of synapses within intact brain tissues by sequentially staining and stripping ultra-thin tissue sections with different antibodies[8,12,13]. More recently, gel embedding and expansion of whole intact organs has been used with sequential antibody loading and stripping to generate 13-channel fluorescence imaging datasets[14]. Cyclic Immunofluorescence (CycIF) was applied to generate 9-channel diffraction-limited images of cancer cell lines using repetitive antibody loading-bleaching cycles[10]. In each case, multiple antibody staining rounds are typically used together with harsh and time-consuming wash-steps that limit both epitope accessibility compared with simultaneous antibody loading, as well as potentially alter epitope reactivity with disruptive chemical or photobleaching treatment. Further, these preceding approaches are not readily amenable to super-resolution imaging within the same intact sample, and are therefore unable to resolve sub-synaptic protein structural organization. While electron microscopy (EM) has been integrated with AT to facilitate correlative light and EM imaging, EM is limited in its ability to resolve multiple molecular species in the same sample[15,16], and requires complex sample fixation, embedding, and processing steps.

In contrast, the use of diffusible fluorescent imaging probes that target specific protein markers or antibodies in situ overcomes the preceding limitations by offering (1) simultaneous antibody loading prior to imaging; (2) rapid probe-exchange using mild buffer treatment and (3) super-resolution imaging using Points Accumulation In Nanoscale Topography (PAINT)[17]. Originally introduced by Sharonov and Hochstrasser[17], PAINT was first used to perform super-resolution imaging of reconstituted lipid membranes with diffusible dye molecules. Subsequently, several variants and applications of this approach[18,19] were introduced to generate multi-channel data, including uPAINT[18], which employs diffusible fluorescent antibodies, and DNA-PAINT[20–24], which uses diffusible fluorescent single-stranded DNA (ssDNA) molecules (imaging probes) that transiently bind to complementary ssDNA oligos (docking strands) attached to target DNA nanostructures[20] or antibodies[21]. Protein fragment-based probes have alternatively been used to generate multiplexed cytoskeletal and focal adhesion super-resolution images with higher labeling density compared with antibody-based approaches[19]. However, this strategy requires identification of highly specific, transiently binding peptides for each target molecular species, which may be challenging to generalize to other proteins, particularly those with lower expression levels than cytoskeletal proteins. Critically, each

of the preceding super-resolution approaches relies on time-consuming and low-throughput time-lapse imaging followed by fluorophore localization and reconstruction employed in the conventional single-molecule localization microscopy approaches PALM[25] and STORM[26]. Moreover, diffusible fluorescent probes typically generate high background fluorescence that prevents their application to quantitative phenotypic profiling of neuronal cell populations using high-throughput confocal imaging of genomic and drug perturbations.

To overcome the preceding limitations in imaging throughput and quantitative phenotyping, here we introduce Probe-based Imaging for Sequential Multiplexing (PRISM). Fluorescently labeled single-stranded locked nucleic acids ssLNA (for LNA-PRISM) and conventional ssDNA (for DNA-PRISM) oligos are employed alternatively as high- versus low-affinity imaging probes to implement either diffraction-limited confocal or PAINT-based super-resolution imaging using the same ssDNA-labeled antibodies or peptide. While in principle confocal imaging can be performed using longer, higher affinity ssDNA imaging probes, as performed previously[22], the design of high specificity and affinity ssLNA imaging probes performed here provides high affinity and specificity yet reversible binding to target docking strands with significantly reduced background fluorescence due to unbound, diffusible imaging probes. This high signal to noise in turn offers quantitative neuronal phenotyping using confocal imaging that was previously impossible using ssDNA probes alone[22,24].

We apply LNA-PRISM to 13-channel confocal imaging of seven synaptic proteins imaged in neuronal culture simultaneously, with five cytoskeletal proteins that have been shown to interact with one another in vivo[27]. These multiplexed imaging data enable the quantitative analysis of 66 protein co-expression profiles extracted from thousands of individual synapses within the same intact neurons, revealing strong correlations amongst subsets of synaptic proteins, as well as heterogeneity in synapse sub-types. In addition, we took advantage of LNA-PRISM to quantify changes in synaptic protein levels, co-expression profiles, and synaptic-subtype compositions in excitatory synapses following blockade of voltage-gated sodium channels with tetrodotoxin (TTX) treatment, as previously performed to examine homeostatic plasticity mechanisms[28–36]. Finally, we apply DNA-PRISM using the same ssDNA-antibody and -peptide conjugates to resolve the 20 nm-scale structural organization of eight synaptic proteins together with filamentous actin and dendritic microtubules in situ. Super-resolution imaging data reveal the nanometer-scale structural organization of nine targets within single synapses that is consistent with EM[37] and average synaptic structure previously assayed using STORM[38].

## Results

**Overview of LNA- and DNA-PRISM.** The PRISM workflow[39] employs neuronal cultures that are fixed, permeabilized, and stained simultaneously using ssDNA-conjugated antibodies or peptides, collectively termed markers, that specifically label cellular targets. Markers are barcoded with single-stranded nucleic acid oligonucleotides (docking strands), rationally designed to optimize orthogonality between complementary fluorescently labeled ssLNA or DNA imaging probes used for confocal or super-resolution imaging (Fig. 1a). To maximize the multiplexing capacity of the assay, primary rather than secondary antibodies are labeled with docking strands whenever possible so that labeling of distinct targets is not limited by the number of secondary antibodies reacting with different species. Extensive validation of each marker and fluorescently labeled ssLNA or ssDNA imaging probe is performed to ensure that markers retain their

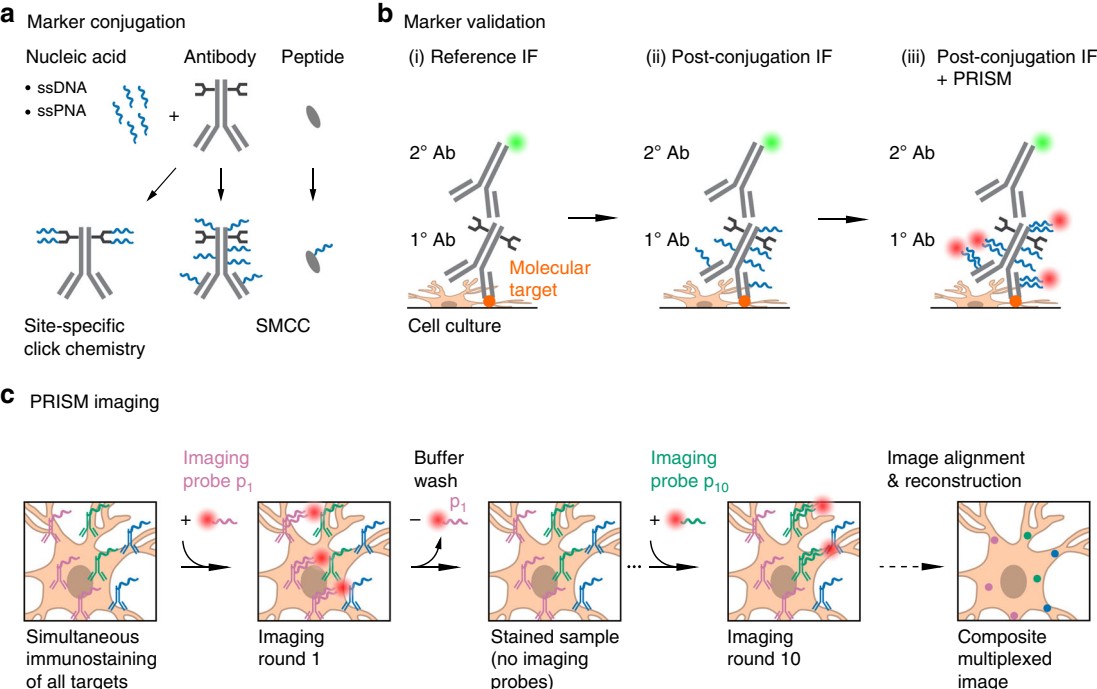

**Fig. 1** Schematic of the PRISM imaging technique. **a** Reagents (markers, shown in gray) for detecting subcellular targets include antibodies or peptides that are conjugated with unique oligonucleotide barcodes (docking strands, shown in blue). A barcoded marker is imaged using the complementary fluorophore-conjugated oligonucleotides (imaging probes) that bind to the docking strands on the marker (see (iii) in **b**, fluorophores shown as red circles). Binding affinity of the imaging probes to the docking strands can be varied by changing the sequence and type of the oligonucleotides, which thereby enables either diffraction-limited (high affinity) or super-resolution microscopy (low affinity). Conjugation of docking strands to markers using site-specific click chemistry enables stoichiometric control of the number of nucleic acids bound to a whole antibody, while SMCC enables conjugation of docking strands to free amine groups on a variety of markers. **b** The reagent testing and validation phase consists of (i) generation of reference staining patterns of all molecular targets of interest using standard immunofluorescence (IF), (ii) analysis of the specificity and staining quality of markers conjugated with docking strands compared to those in the reference IF, and (iii) co-localization of PRISM-imaged staining patterns using imaging probes (red circles, which correspond to fluorophores conjugated to the probes) with standard IF staining patterns (green circles). **c** Overview of the main steps in the PRISM imaging workflow. All molecular targets of interest are immunostained at once using docking strand-conjugated markers (e.g., antibodies shown in green, blue, and pink). Nucleic acid imaging probes specific to each marker (e.g., $p_1$–$p_{10}$) are applied and imaged sequentially, with each imaging strand washed out after image acquisition at each step. This approach enables imaging a dozen or more distinct molecular targets in the same sample. Images of different markers are drift-corrected and overlaid to generate a pseudo-colored, multiplexed image. For super-resolution PRISM, prior to drift correction, the super-resolved image of each marker is reconstructed from the temporal image stack of binding/unbinding events of the imaging probes to/from the docking strands on the marker

target-specific recognition properties following conjugation with nucleic acid docking strands and that imaging strands target cognate docking strands with high affinity and specificity without cross-talk (Fig. 1b). Following marker and imaging probe validation, multiplexed imaging is performed using sequential labeling and washing out of individual imaging probes, with wash-steps in between used to clear the sample of imaging probes (Fig. 1c). Diffraction-limited, confocal images are acquired with LNA-PRISM, whereas single-molecule time-lapse imaging followed by image reconstruction, drift correction, and image alignment is performed with DNA-PRISM using PAINT.

**Design and validation of markers for PRISM.** To apply multiplexed neuronal imaging in either confocal or super-resolution modes using the same protein markers, we conjugated distinct ssDNA docking strands using either SMCC linkers or site-specific chemoenzymatic labeling to synaptic and cytoskeletal markers validated in neuronal cell culture (Supplementary Fig. 1, Supplementary Note 1 and Methods). Whereas SMCC conjugates docking strands to surface-accessible primary amines through NHS chemistry, site-specific labeling conjugates ssDNA docking strands to four conserved glycan chains on the Fc region of the antibody (Supplementary Figs. 2 and 3), thereby minimizing the likelihood of disrupting the antibody paratope. In generating PRISM antibodies, SMCC was first attempted for ssDNA conjugation, and site-specific conjugation was alternatively applied to antibodies that changed localization patterns after SMCC-based conjugation. In cases where staining patterns were disrupted with both SMCC and site-specific conjugation or the fluorescence signal of the primary antibody conjugate was too low for high quality imaging, ssDNA-conjugated secondary antibodies were instead employed to visualize the target.

Because conjugation of antibodies and peptides with ssDNA may alter their affinity and/or specificity, we validated each marker-docking-strand conjugate individually in neuronal culture using indirect immunofluorescence (IF) to ensure the same staining patterns were obtained compared with the reference, unconjugated marker. However, most markers were found to exhibit strong nuclear localization following SMCC or site-specific conjugation with ssDNA, suggesting that the observed change in affinity of the ssDNA-conjugated antibody to its target is not solely due to the possible modification of paratopes by ssDNA (Fig. 2a, Supplementary Figs. 4 and 5).

To eliminate off-target nuclear localization of the ssDNA-conjugated antibodies, we screened several nuclear blocking

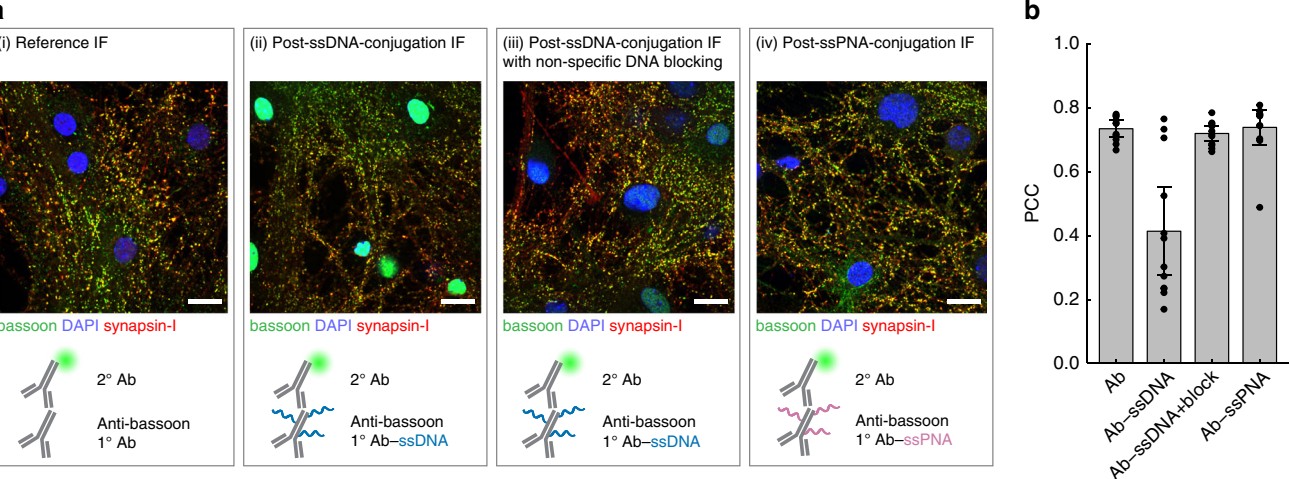

**Fig. 2** Blocking off-target nuclear localization of ssDNA-conjugated antibodies. **a** Neurons were stained with native or ssDNA-conjugated anti-bassoon antibody, anti-synapsin-I antibody, and DAPI. ssDNA-conjugated anti-bassoon antibody exhibited strong off-target nuclear localization (ii, green staining inside the nuclei), compared to the native antibody (i). This nuclear localization was reduced by blocking the fixed sample with non-specific (salmon sperm) DNA prior to immunostaining (iii) or when the anti-bassoon antibody used for staining was conjugated with ssPNA instead of ssDNA (iv). Scale bars: 20 μm. **b** Cross-correlation analysis of the IF images in **a**. Pearson correlation coefficient (PCC) of the bassoon channel (green in **a**) with the synapsin-I channel (red in **a**) for each image. Differences in PCC indicate changes in antibody staining patterns. Error bars represent 95% confidence intervals

agents and found that salmon sperm DNA commonly used in Southern blotting successfully blocked the nuclear localization of ssDNA-conjugated antibodies (Fig. 2a and Supplementary Fig. 6). Interestingly, conjugating antibodies with single-stranded Peptide Nucleic Acid (ssPNA)[40] docking strands instead of ssDNA also eliminated nuclear localization in the absence of any blocking (Fig. 2a), supporting the hypothesis that overall charge of the nucleic acid docking strands present on antibodies was responsible for their non-specific nuclear localization. Nevertheless, salmon sperm blocking was performed in all experiments to minimize cost and complexity associated with generating a full library of ssPNA docking strands. Image cross-correlation analysis showed that ssDNA-conjugated antibodies produced staining patterns similar to those of unmodified antibodies, as assessed through indirect IF when samples were blocked with salmon sperm DNA prior to immunostaining (Fig. 2b and Supplementary Fig. 7).

**Imaging probe design for LNA-PRISM.** To enable high-throughput confocal imaging of neurons with high signal-to-noise and low background fluorescence in multi-well plate format, we designed high affinity ssLNA imaging strands of 11 nt length that target the same 11 nt ssDNA docking strands used in DNA-PAINT imaging with high specificity[21]. Similar to ssDNA, ssLNA binding affinity to complementary ssDNA is salt-dependent, thereby enabling rapid probe exchange via imaging probe wash-out using low salt concentration buffer (Supplementary Fig. 8). RNase was used to eliminate off-target binding of ssLNA imaging probes to cellular RNA (Supplementary Fig. 9), a treatment that did not affect antibody marker localization (Supplementary Fig. 10). Orthogonality of each imaging probe was validated individually using a cell-based crosstalk assay that resembles the staining and imaging conditions in a multiplexed PRISM experiment. Results of this cross-talk assay showed <10% crosstalk between each of the docking-imaging-strand pairs for our staining and imaging conditions (Supplementary Figs. 11, 12 and Supplementary Note 2). Typical imaging strand incubation and wash-out times for LNA-PRISM are 5–10 min each, which is considerably faster than existing multiplexed imaging approaches that require multiple rounds of antibody staining and elution that

can require up to hours or days to complete[8,10,12]. PRISM washing conditions consist of 0.01x phosphate buffered saline (PBS), which is also milder than alternative multiplexed imaging methods that utilize oxidizing reagents or high-pH buffer[8,10,12]. In addition to reducing the risk of altering epitopes over the course of multiple wash cycles, mild buffer conditions minimize the possibility of the disruption of delicate cellular structures, which may be particularly crucial for preserving the integrity of neuronal synapses. RNase-treated cells produced target staining patterns using LNA-PRISM that were indistinguishable from those with conventional indirect IF (Supplementary Fig. 13).

**Quantitative multi-channel confocal neuronal imaging.** 13-channel imaging of cultured rat hippocampal neurons using ten ssLNA imaging probes and three non-PRISM fluorescent markers was performed to characterize the synaptic and cytoskeletal protein–protein network that is core to the regulation of synapse formation and plasticity (Fig. 3a). This network includes the cytoskeletal proteins actin, Tuj-1, MAP2, ARPC2, and cortactin, the pre-synaptic proteins synapsin-I, bassoon, and VGLUT1, the post-synaptic density proteins PSD-95, Homer-1b/c, and SHANK3, and the NMDA-type glutamate receptor subunit NR2B. The canonical synaptic markers synapsin-I, bassoon, VGLUT1, PSD-95, Homer-1b/c, and SHANK3 exhibited a high degree of co-localization, with punctate patterns, whereas Tuj-1 and MAP2 yielded clear cytoskeletal morphologies (Fig. 3b and Supplementary Fig. 14). Noticeably, ARPC2 and cortactin displayed punctate patterns that also co-localized with other synaptic markers, in agreement with previous results[27,41,42]. To assess whether multiple rounds of imaging probe wash-out and probe application steps physically distorted the sample or noticeably stripped markers from their epitopes, synapsin-I was imaged twice, once in the middle and once at the end of the PRISM experiment, which revealed highly reproducible localization patterns (Fig. 3a).

LNA-PRISM offers nearly an order of magnitude increase in the ability to detect co-localization/co-expression patterns in situ, with 66 protein–protein co-localizations using 12 protein labels, compared with only six from conventional four-channel imaging. While synaptic proteins generally showed co-localization,

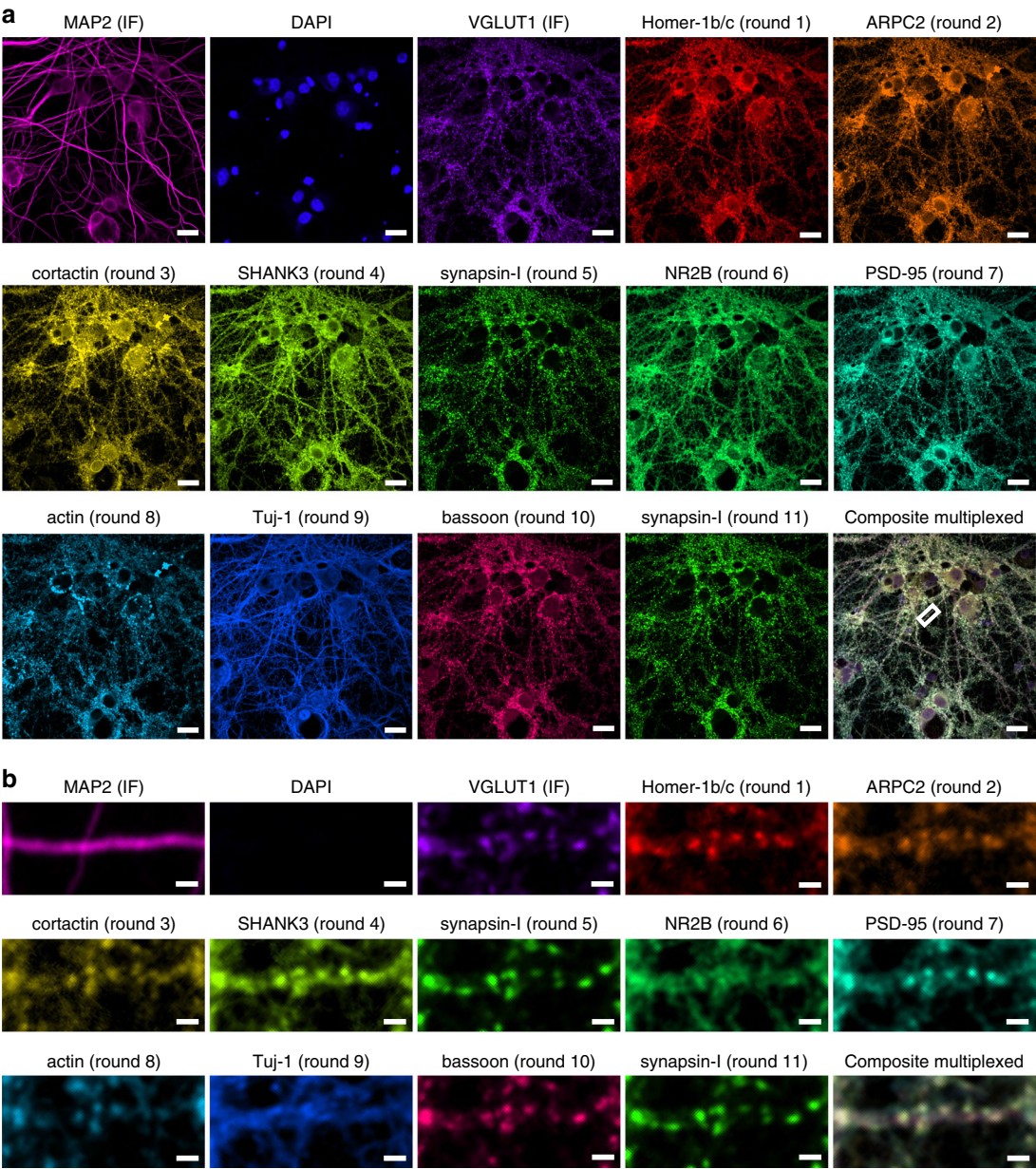

**Fig. 3** Confocal LNA-PRISM images of rat hippocampal neuronal synapses. **a** 13-channel images of 21 days in vitro (DIV) rat hippocampal neuronal culture from a single field of view. Individual channels for each marker are shown followed by a composite image in the bottom-right corner. MAP2 and VGLUT1 were visualized using fluorescently labeled secondary antibodies, nuclei were visualized using DAPI, and all other targets were visualized using ssLNA imaging probes. Synapsin-I was imaged twice, once in the middle and once at the end of the experiment. **b** Zoom-in view of a single dendrite indicated by the white box in **a**. Scale bars: **a** 20 μm; **b** 2 μm

examination of individual synapses revealed variations in co-localization patterns across different proteins and synapses (Fig. 4a). To characterize the co-localization/co-expression patterns of each protein pair, individual synaptic features, including size and intensity, were extracted for each target from PRISM images using an image-processing pipeline optimized for synapse segmentation (Supplementary Figs. 15, 16 and Methods). Correlations between distinct synaptic features were computed across all synapses, with a high correlation score between two synaptic proteins indicating a higher functional association. The correlation scores showed expression levels of most synaptic proteins were highly correlated in synapses, with the exception of the cytoskeletal proteins Tuj-1 and MAP2, in agreement with previous image cross-correlation analyses that showed that tubulin is largely excluded from synapses[12] (Fig. 4b and

Supplementary Fig. 17). In addition, the post-synaptic density proteins Homer-1b/c, PSD-95, and SHANK3 strongly correlated with one another in their expression levels, which may be attributed to their dense and compact protein distributions within the post-synaptic density[43,44]. The ARP2/3 complex subunit ARPC2, which has been shown to interact with SHANK3 within synapses[27], also correlated in its expression level with other post-synaptic density proteins. In agreement with the expected separation of pre- and post-synaptic proteins, synapsin-I and VGLUT1, which are associated with pre-synaptic vesicles, were highly correlated with the pre-synaptic scaffolding protein bassoon, but were only weakly correlated with most post-synaptic proteins (Fig. 4b). Interestingly, bassoon and VGLUT1 exhibited moderate correlation in expression with PSD-95, suggesting a coordination of pre- and post-synaptic structures

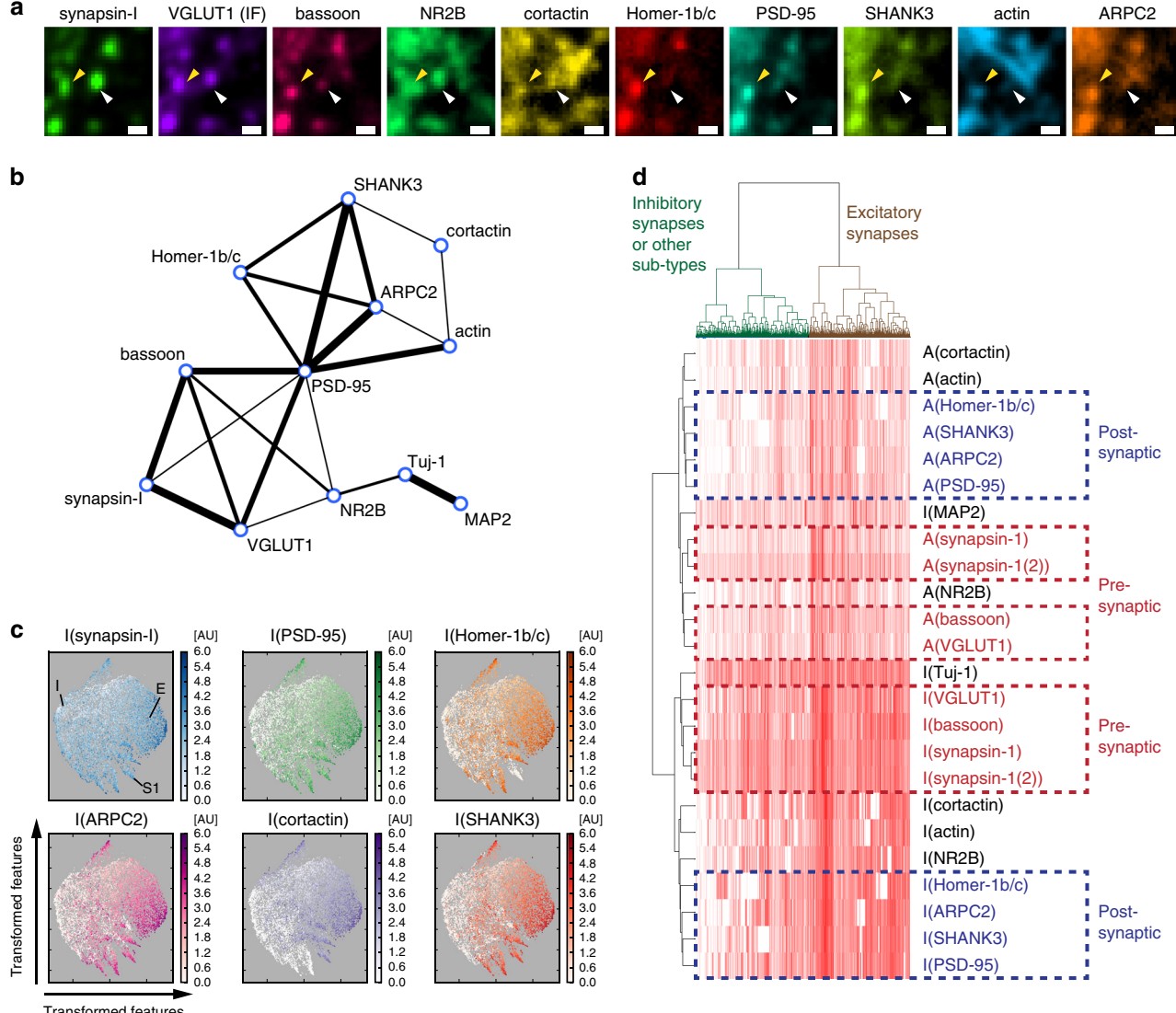

**Fig. 4** Analysis of single-synapse profiles from confocal imaging. **a** LNA-PRISM images of a conventional excitatory synapse (yellow arrowhead) with co-localization of every synaptic marker measured, and a synapse with only a subset of markers present (white arrowhead). Scale bar: 1 μm. **b** Network representation of correlations between intensity levels of synaptic proteins within synapses ($n = 178,528$ synapses from three cell culture batches). The thickness of each edge represents the relative correlation strength between the respective nodes. **c** t-Distributed Stochastic Neighbor Embedding (t-SNE) plots of $n = 10,000$ synapses from a single culture batch; each with 20 features (intensity levels and punctae sizes of ten synaptic proteins). Each point in each t-SNE map represents a single synapse with its $(x,y)$ coordinates corresponding to the transformed features that best preserve the distribution of synapses in the original high dimensional feature space. Intensity levels of individual proteins are color-coded in each map. E: cluster of conventional excitatory synapses with the presence of most synaptic markers; I: cluster of possible inhibitory synapses with the absence of most synaptic markers; S: cluster of possible sub-type synapses with the presence of only a subset of synaptic markers. **d** Hierarchical clustering analysis of synapse profiles. Each column in the heat map represents a profile of a single synapse with 24 synaptic features (rows). I and A denote image intensity level and punctum size, respectively ($n = 53,698$ synapses from a single culture batch)

across the synaptic cleft. NR2B exhibited correlation with both pre- and post-synaptic markers, consistent with several previous observations of both pre- and post-synaptic localizations of NMDAR[45,46].

High-throughput, large-scale profiling of synapses also enabled us to use the rich protein co-expression feature profiles assayed with LNA-PRISM to classify synapse sub-types using machine learning. To identify putative sub-categories of synapse types in this high-dimensional feature space, consisting of 24 dimensions, we applied t-Distributed Stochastic Neighbor Embedding (t-SNE), a tool commonly used to visualize high-dimensional single-cell data (Fig. 4c)[47]. t-SNE transforms high dimensional data into two dimensions, aiming to preserve the local high dimensional

data structure within the lower dimensional space. t-SNE analysis of 10,000 randomly subsampled synaptic profiles revealed a cluster of synapses contain most of the synaptic proteins that we measured, which, given that our antibody panel consisted mostly of excitatory synapse-associated proteins, also likely corresponded to conventional excitatory synapses. In addition, smaller sub-type clusters were identified, showing an absence of one or more synaptic proteins, which may correspond to conventional inhibitory synapses or additional synapse sub-types. Hierarchical clustering of protein feature profiles corroborated findings of the preceding correlation and t-SNE analyses, namely that pre-synaptic proteins are highly clustered with one another, whereas post-synaptic density proteins and ARPC2 form a separate sub-

cluster (Fig. 4d). Features of synapsin-I from two imaging rounds were tightly clustered together, confirming that clustering results reflected the similarities between features. These findings suggest that protein associations derived from PRISM data recapitulate the molecular composition and structural properties of excitatory synapses and can do so for one dozen targets simultaneously in thousands of synapses within the same intact sample imaged within several hours in multi-well plate format.

**Synaptic protein reorganization following activity blockade.** To test the ability of LNA-PRISM to characterize activity-dependent changes in excitatory synapses under homeostatic plasticity, hippocampal neurons were treated for 48 h with TTX to chronically reduce overall neuronal activity levels and thereby induce scale-up of synaptic strength. Previous studies have used two-color imaging, western-blotting, and electrophysiology to characterize changes in synaptic protein levels associated with neurotransmission in response to chronic network activity inhibition[28–30,33,35,36,48–50]. However, these studies either averaged over individual synapses by analyzing bulk protein levels biochemically or visualized in situ only a small subset of synaptic markers due to limitations in conventional fluorescence imaging. In contrast, here we characterize the levels of nine synaptic proteins simultaneously within individual synapses following induction of homeostatic plasticity. We measure integrated synapse intensity and synapse area for each synaptic protein target located within 1 μm of synapses identified using a synapsin-I mask (Fig. 5a and Supplementary Fig. 18).

Following treatment, we observe a significant increase in overall integrated intensity of the pre-synaptic proteins VGLUT1 and bassoon, and the post-synaptic proteins Homer-1b/c, PSD-95, and SHANK3 (Fig. 5b and Supplementary Data 1). In contrast, chronic treatment with TTX did not induce significant changes in the integrated intensity of cortactin, actin, NR2B, or synapsin-I at the 0.05 p-value reported using a two-tailed Student's *t*-test. Analysis of punctae areas shows significant changes in all of the synaptic scaffolding proteins evaluated in this study, corroborating a robust increase in the levels of these proteins present at synapses (Fig. 5b). Mean synaptic intensities were also computed, with only VGLUT1 and SHANK3 showing significant changes (Supplementary Fig. 19). This indicates that for SHANK3 the integrated intensity increases to a greater degree than the area following TTX treatment, whereas for bassoon, Homer-1b/c, and PSD-95 the increases in areas are similar to the increases in intensities.

Compared with conventional imaging, which also offers the ability to profile changes in individual synaptic protein levels across distinct samples, PRISM uniquely offers the ability to simultaneously monitor changes in numerous synaptic protein levels within the same intact sample. Pair-wise correlation analysis of synaptic markers revealed correlation coefficients for some pairs of synaptic proteins that were distinct from previous replicates, as expected due to typical variations in neuronal culture conditions (Supplementary Figs. 20 and 21). Notwithstanding, our analysis revealed reproducible correlations between Homer-1b/c:SHANK3, Homer-1b/c:PSD-95, SHANK3:PSD-95 and VGLUT1:bassoon (Supplementary Figs. 17 and 21). Following TTX treatment, increased correlation was observed between the active zone marker bassoon and each of the synaptic vesicle zone markers VGLUT1 and synapsin-I (Supplementary Fig. 20 and Supplementary Data 2). Notably, despite overall increases in integrated intensity and area, no significant changes were detected in the associations amongst the post-synaptic density markers Homer-1b/c, SHANK3, and PSD-95, consistent with previous work that suggests groups of synaptic proteins are co-

regulated.[28,30] Taken together, this suggests that there is a change in the stoichiometry of the pre-synaptic, but not the post-synaptic components. To assess whether activity blockade induces changes in association between pre- and post-synaptic compartments, correlation analysis of pre- and post-synaptic proteins was performed. Increased association between post-synaptic density zone markers and synaptic vesicle zone markers is evident due to the significant correlation between the protein pairs synapsin-I: Homer1b/c, VGLUT1:SHANK3, and VGLUT1:PSD-95 in TTX-treated samples. In contrast, a decrease in the association of cortactin with the post-synaptic density markers Homer-1b/c and PSD-95 was observed, suggesting a possible redistribution of cortactin from the post-synaptic density region closest to the post-synaptic membrane following TTX treatment (Supplementary Fig. 20 and Supplementary Data 2).

t-SNE analysis applied to 20,000 neuronal synapses ($n = 10,000$ from each treatment group, $n = 2000$ synapses from each untreated or treated replicate) offered the opportunity to identify categories of synapse sub-types associated with network activity blockade that would otherwise be impossible to identify using conventional two- or three-color imaging (Fig. 5c–e). We used kernel density estimation to identify interesting features in the t-SNE analyses which identified at least seven potential synaptic sub-categories (Fig. 5c). While each of these regions is present in the treated and untreated groups (Fig. 5d and Supplementary Fig. 22), in the TTX-treated group the features appeared shifted towards regions of higher protein levels and larger punctae areas (Supplementary Figs. 12–31). Examination of the individual t-SNE plots to characterize the different regions using the presence of excitatory markers (Fig. 5e and Supplementary Figs. 24, 26, 28-31) revealed that Regions 1–6 are varying sub-classes of excitatory synapses, whereas Region 7 contains either inhibitory synapses based on the absence of excitatory markers (Supplementary Figs. 24, 26, 28–31), or non-synaptic synapsin-I punctae (Supplementary Figs. 23–25).

Examination of the individual t-SNE plots reveals that Region 1 is comprised of synapses with synapsin-I and VGLUT1, but low in bassoon and other synaptic markers (Supplementary Figs. 23–31). These synapses may either be immature excitatory synapses or non-synaptic clusters of pre-synaptic proteins. Region 2 contains excitatory synapses with low PSD-95 levels (Supplementary Fig. 29), whereas Region 3 consists of excitatory synapses with low actin levels (Supplementary Fig. 31). Regions 4 and 5 consist of excitatory synapses with all synaptic markers present, however, Region 4 contains medium intensity punctae whereas Region 5 has high intensity punctae (Fig. 5e). Region 6 also contains excitatory synapses, yet with lower levels of cortactin (Supplementary Fig. 27).

While we describe seven putative categories of synapse sub-types here, there are likely additional sub-categories present that can be characterized using different combinations of synaptic markers. Because it is currently unknown how many distinct synaptic sub-types are present in neurons, how these synapses vary across different cell-types and brain regions, and whether there are functional differences between these sub-types, PRISM offers an important tool to explore these questions.

**Super-resolution neuronal imaging using DNA-PAINT.** The same antibody-ssDNA conjugates offered the ability to also super-resolve molecular targets within individual synapses in primary mouse neuronal cultures using DNA-PRISM (DNA-PAINT)[20,21]. Neuronal cultures were assembled into flow cells in which fluid exchange was controlled by an automated fluidics handling system to ensure gentle buffer washing and imaging probe application designed to minimize sample distortion. Super-

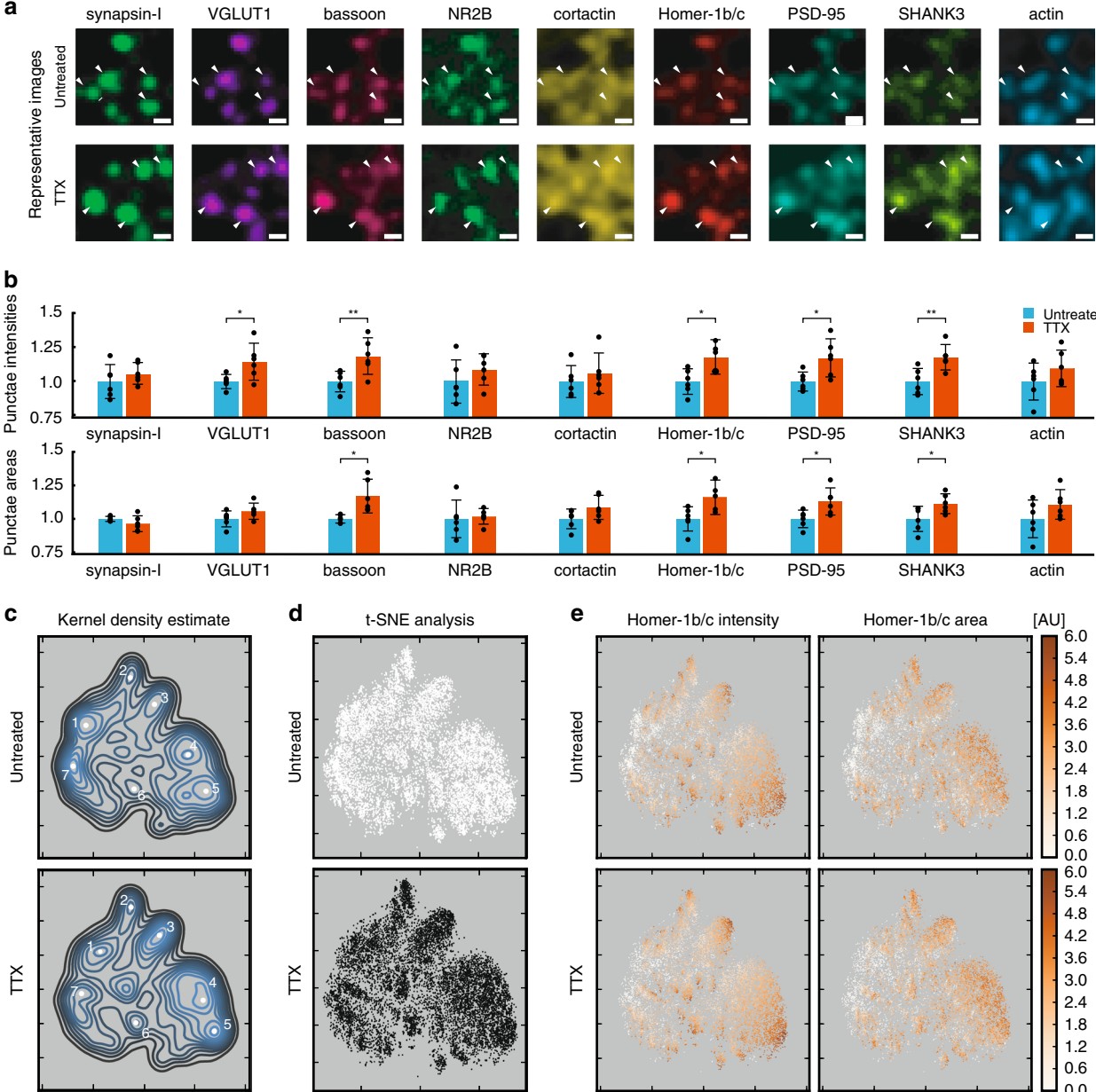

**Fig. 5** Analysis of synaptic remodeling from confocal imaging. **a** Representative images of synapses from 21 days in vitro rat hippocampal neurons. White arrowheads indicate synapses. Scale bar: 1 μm. **b** Bar heights indicate the average synaptic integrated intensities (top) or average synaptic punctae areas (bottom) of untreated and TTX treated neurons relative to the mean of the untreated group. Error bars represent 95% confidence intervals. *P*-values are computed using Student's *t*-test with *n* = 6 replicates; *p* < 0.05 (*), *p* < 0.01 (***). **c–e** t-SNE plots of individual synapses from untreated (*n* = 10,000) and TTX treated (*n* = 10,000) synapses. Mean intensities and areas from each target are used as input. **c** Kernel density estimate of t-SNE output. White points indicate potential synaptic sub-types. **d** Individual points color-coded to indicate untreated (white) or TTX treated (black) synapses. **e** Individual points are color-coded for Homer-1b/c mean intensity and Homer-1b/c area

resolution DNA-PRISM images of microtubules and actin in neurons were first compared with widefield IF images. Super-resolved microtubules formed bundles within neuronal processes, whereas actin exhibited linear, filamentous structures within these regions, but showed punctate structures within dendritic spines (Fig. 6a–d). Subcellular structures imaged with DNA-PRISM correlated well with the corresponding widefield images, but with significantly improved spatial resolution (Fig. 6b, d). A Gaussian fit to the cross-sectional profile of a single microtubule produced a full-width at half-maximum (FWHM) of 46.5 nm (Fig. 6g), which is consistent with previous PAINT measurements in HeLa cells[21]. In addition to microtubules and actin, DNA-PRISM

imaging of neuronal synapses also corresponded well with the widefield IF images, but with significantly improved resolution, as expected, revealing closely apposed pre- and post-synaptic sites (Fig. 6e–f). We quantified the average synapse size defined by synapsin-I and PSD-95 punctae using the radial cross-correlation function between synapsin-I and PSD-95, with the decay length of the correlation function revealing an average synapse size of ~200 nm (Fig. 6i)[12]. The spatial decay of the DNA-PRISM correlation curve occurred at a smaller spatial scale than the widefield imaging curve, indicating the smaller synapse size revealed by DNA-PRISM due to enhanced resolution relative to widefield imaging.

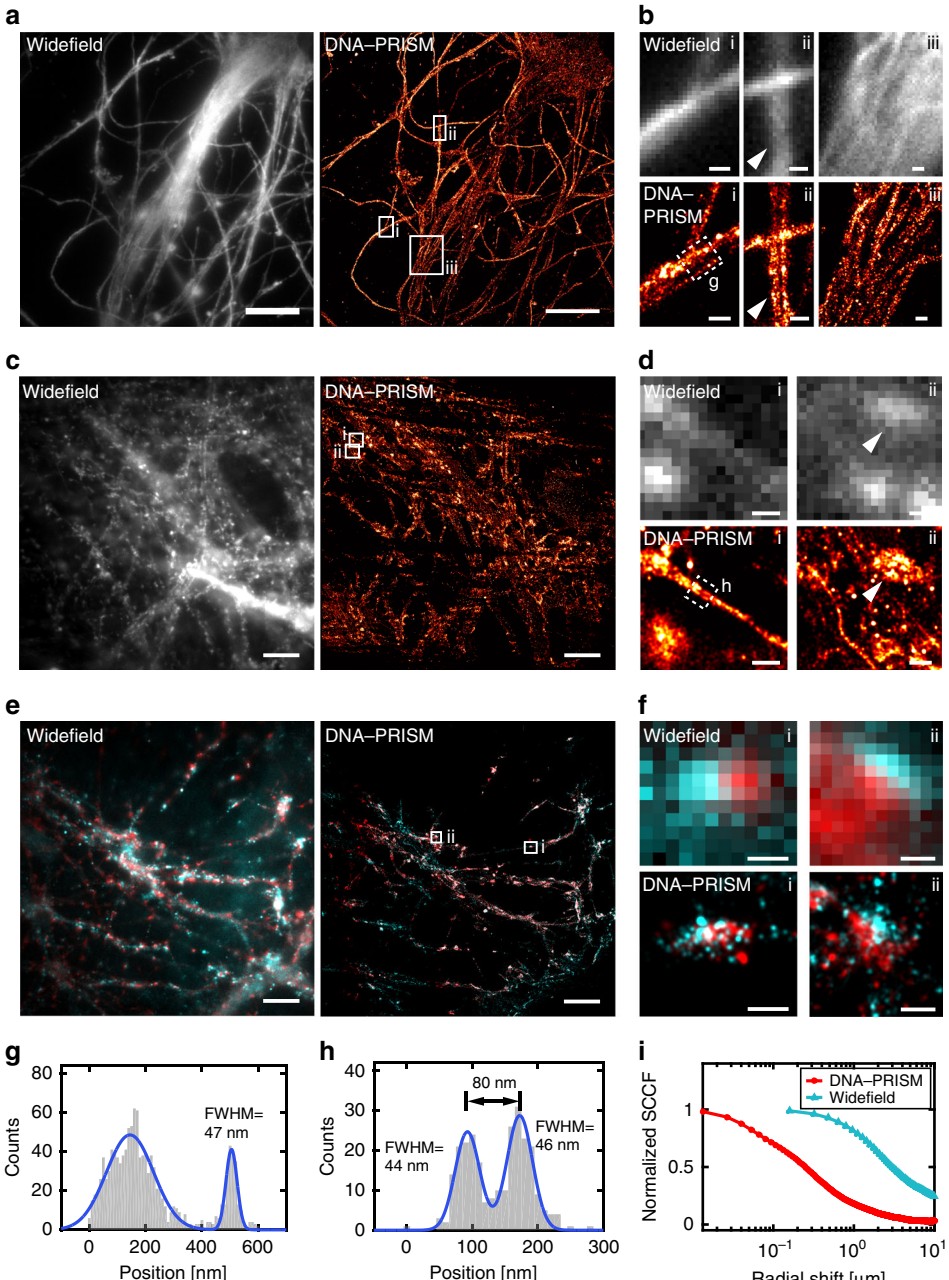

**Fig. 6** Super-resolution DNA-PRISM imaging of primary neuronal cultures. **a** Widefield and DNA-PRISM images of neuronal microtubules stained using the DNA-conjugated anti-Tuj-1 antibody. **b** Zoom-in view of the boxed areas in (**a**) show resolution enhancement of DNA-PRISM images compared with widefield images. The arrowhead indicates distinct microtubule bundles that are not resolved in the widefield image. **c** Widefield and DNA-PRISM images of filamentous actin stained using DNA-conjugated phalloidin. **d** Zoom-in views of the boxed areas in **c** show two actin filaments (left) and the synaptic actin punctae with sub-synaptic structures (right, arrowhead) that are not resolved in widefield images. **e** Widefield and DNA-PRISM images of pre-synaptic marker synapsin-I (red) and post-synaptic marker PSD-95 (cyan) of the same field of view. **f** Zoom-in view of single synapses indicated by boxes in **e**. **g** Cross-sectional profile of the boxed region in (**b**) shows a microtubule bundle next to a possible single microtubule with FWHM = 47 nm. **h** Cross-sectional profile of the boxed region in **d** shows two actin filaments or small filament bundles that are 80 nm apart. **i** The average size of synapses defined by synapsin-I and PSD-95 is quantified using the normalized radial cross-correlation function. The decay at the smaller radial shift of the DNA-PRISM curve (red) indicates the smaller synapse size in the DNA-PRISM image due to the improved spatial resolution. Scale bar: **a**, **c**, **e** 10 μm; **b**, **d**, **f** 0.5 μm

We next applied DNA-PRISM imaging to super-resolve the nanoscale pre- and post-synaptic organization of 9 targets within individual synapses, including Tuj-1, actin, cortactin, PSD-95, synapsin-I, NR2B, SHANK3, Homer-1b/c, and bassoon (Fig. 7a). Due to differences in synapse orientations with respect to the imaging plane, individual synapses varied in their degree of overlap among proteins within each synapse. For a subset of synapses with the proper orientation relative to the imaging

plane, we identified clear separation between pre-synaptic proteins (synapsin-I and bassoon) and post-synaptic proteins (PSD-95, SHANK3, Homer-1b/c) (Fig. 7b). Pre- and post-synaptic proteins were localized to opposing regions of the synaptic cleft, cytoskeletal proteins (Tuj-1, actin, cortactin) and NR2B were observed in both sides of the cleft. Moreover, post-synaptic density proteins (PSD-95, SHANK3, Homer-1b/c) showed narrow, overlapping distributions in expression levels,

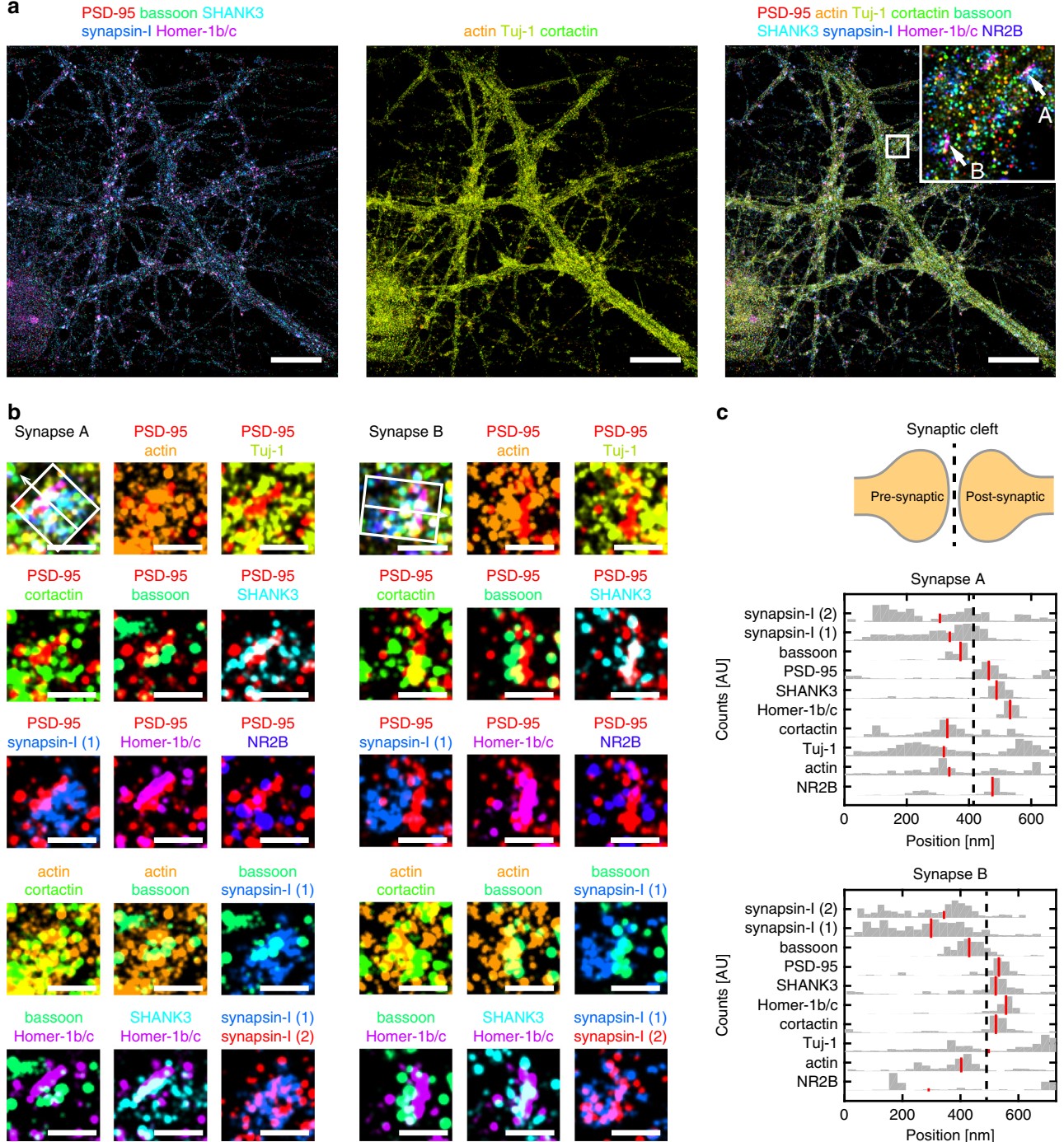

**Fig. 7** Distributions of synaptic proteins within individual synapses. **a** DNA-PRISM images of neurons showing the subset channels of synaptic proteins (left), the subset channels of cytoskeletal proteins (middle), and all the channels (right). **b** Zoom-in view of two individual synapses in **a** shows the separation of pre-synaptic proteins (synapsin-I and bassoon) and post-synaptic proteins (PSD-95, SHANK3, Homer-1b/c). For each synapse, the nine-target image is shown in the top-left corner, with distinct pairs of synaptic proteins shown in the remaining images. Synapsin-I was imaged twice, once at the beginning and once at the end of the experiment. **c** Cross-sectional profiles of protein distributions along trans-synaptic axes (white boxes with arrows in **b**) of the two synapses in **b**. Red lines indicate the medians of the distributions. Scale bars: 10 μm in full field views; 500 nm in zoom-in views

suggesting physical interaction of these proteins in the post-synaptic density (Fig. 7c) that is consistent with the correlation analysis applied to the preceding confocal imaging results. In contrast, synapsin-I exhibited a broader spatial distribution compared with the distributions of scaffolding proteins, in agreement with the more diffuse distributions expected for vesicle-associated proteins (Fig. 7c). These spatial distributions of synaptic proteins were consistent with the average distributions

previously measured from multiple synapses and distinct cultures using three-channel STORM[38] and EM[37]. However, in stark contrast to these previous studies that relied on reference markers, our imaging and analysis of sub-synaptic proteins resolved all measured targets of interest within the same synapse simultaneously. Integration of our approach with 3D super-resolution imaging systems would offer its application to dozens or hundreds of synapses in situ.

## Discussion

LNA-PRISM uses diffusible, high-affinity ssLNA imaging probes to realize high-throughput confocal imaging for rapid, large-scale quantitative phenotypic screening of more than one-dozen cytoskeletal and synaptic protein targets across tens of thousands of individual neuronal synapses under different conditions. Alternately, DNA-PRISM utilizes the same antibody/peptide reagents with ssDNA imaging probes to perform super-resolution synaptic imaging with PAINT. In future studies, large-scale morphological screens may first be performed in multi-well plate format using LNA-PRISM, followed by super-resolution imaging of a sub-set of synapses or neuronal sub-regions with DNA-PRISM to resolve synaptic ultra-structure. Compared with previous multiplexed diffraction-limited imaging approaches that utilize sequential antibody labeling and stripping or bleaching[8,10,12], LNA-PRISM offers simultaneous staining of all protein targets, which reduces the risk of masking antigens, as well as substantially increasing assay throughput. The use of physiological wash buffers additionally minimizes the possibility of sample or epitope disruption, which may be crucial for high-resolution structural and co-localization studies requiring high sample fidelity, such as in the profiling of synapses within cultured neuronal samples. Application of our primary antibody conjugation strategy together with the use of libraries of orthogonal ssDNA sequences[51] offers the potential for future application of PRISM to neuronal and other cellular systems exceeding substantially the approximately one dozen targets realized here. Genetic perturbation screens aimed at discovering subtle alterations in neuronal phenotype should additionally benefit substantially from the large-scale protein association networks within synapses that are derived from the 12 synaptic targets examined in this study, which offer 66 pair-wise synaptic co-localizations within the same neuronal culture, as demonstrated here for TTX-perturbation that impacts neuronal synapse plasticity.

Using PRISM, we concurrently examine neuronal response to TTX treatment for nine synaptic targets at individual synapses. Our results are consistent with findings described previously, demonstrating increases in post-synaptic scaffolding proteins such as PSD-95, Homer-1b/c and SHANK3 during TTX-induced homeostatic scaling[28,30,33,35,50]. Less is known about the pre-synaptic response during homeostatic synaptic plasticity, predominantly that TTX treatment increases calcium influx in response to action potentials, and also increases the probability of vesicle fusion[52]. Consistent with previous work,[28,34,35] we find increases in both bassoon and VGLUT1 but not synapsin-I, which suggests the pre-synaptic changes may be restricted to the active zone. Thus, PRISM offers a powerful approach to multiplexed fluorescence imaging of cellular protein targets in situ for both phenotypic profiling and high-resolution structural analysis of neuronal cultures.

## Methods

**SMCC ssDNA conjugation of antibodies and phalloidin.** Twenty-five nmol of thiolated single-stranded DNA (ssDNA) (Integrated DNA Technologies) was reduced using 50 mM Dithiothreitol (DTT) for 2 h, purified using a NAP-5 column (GE Healthcare Life Sciences), and quantified using a NanoDrop 2000 (Thermo Fisher Scientific). In parallel, 100 µg of antibody was concentrated to 1 mg/mL using a 100 kDa Amicon Ultra 0.5 mL centrifugal column (EMD Millipore) and purified from additive chemicals such as sodium azide using a 7 kDa Zeba spin desalting column (Thermo Fisher Scientific). A freshly prepared solution of SMCC with 5% DMF (Sigma-Aldrich) in PBS was then added to react for 1.5 h with the antibody solution at a molar ratio of 7.5:1. Unreacted SMCC (succinimidyl 4-(N-maleimidomethyl)cyclohexane-1-carboxylate) was removed using a 7 kDa Zeba column. In a subsequent reaction, antibodies were mixed in a 1:15 molar ratio with the reduced thiolated ssDNA strands and incubated overnight at 4 °C to form stable thioether bonds. ssDNA-conjugated antibodies were then purified using a 100 kDa Amicon Ultra 0.5 mL column. The final protein concentration of the antibody was measured using a NanoDrop 2000 and the antibodies were stored at −20 °C in PBS with 50% glycerol. Amino-modified phalloidin (Bachem AG) was

conjugated using the protocol described above with an extra step of HPLC purification to separate unreacted thiolated ssDNA and ssDNA-conjugated peptides. The optimal antibody concentration was empirically determined for each different conjugation batch. See Supplementary Tables 1 and 3 for antibody information.

**Site-specific ssDNA conjugation of antibodies.** The SiteClick antibody labeling system (Thermo Fisher Scientific) enables site-specific conjugation to four conserved glycan sites present on the Fc region of the heavy chains using copper-free click chemistry. Briefly, 100 µg of each antibody was concentrated to 2–4 mg/mL in azide-free Tris buffer and treated with β-galactosidase enzyme to modify carbohydrate domains. In a second step, azide modified sugars were attached to the modified glycan chain using β-1,4-galactosyltransferase. After overnight incubation and purification of antibodies using a 100 kDa Amicon Ultra 0.5 mL centrifugal column, DBCO-modified ssDNA docking strands (Integrated DNA) were mixed at a 40:1 molar ratio with azide-modified antibodies and incubated overnight at 25 °C. ssDNA conjugated antibodies were then purified using a 100 kDa Amicon Ultra 0.5 mL. Final concentration was measured using a NanoDrop 2000 and antibodies were stored in PBS with 50% glycerol. For PNA-antibody conjugation, DBCO-modified mini-PEG-γ-PNA strands[40] (PNA Innovations) were used as the docking strands. See Supplementary Tables 1 and 3 for antibody information.

**Design of ssLNA imaging probes.** Same ssDNA sequences used for DNA-PAINT were used for ssLNA imaging probes with two modifications to increase the melting temperatures of the sequences: (1) two adenine nucleotides were added to the 3′ end of each sequence of the imaging probes to maximize the hybridization of the imaging probes with the corresponding docking strands. (2) Three DNA nucleotides in each sequence were replaced with LNA nucleotides. Since introducing LNA nucleotides could possibly introduce unwanted cross-hybridization (or crosstalk) between docking strands and imaging probes, an empirical algorithm was used to determine the optimal positions to replace the DNA nucleotides. Briefly, BLAST was performed for each docking strand and imaging probe pair to identify the partially complementary regions for each pair. An empirical cost function was then used to calculate the cost of every possible LNA substitution scheme with each docking strand. The cost function penalized LNA substitution in the partially complementary regions, with increasing penalty for more LNA substitution in the partially complementary regions, and bigger partially complementary regions. For each LNA substitution scheme, the highest cost within all the costs with different docking strands was assigned as the cost for the given substitution scheme. Finally, for each LNA sequence, the substitution scheme with the minimal cost was then selected for our LNA imaging probe sequence design. The algorithm was implemented in Python 3.5. Blast was performed using Biopython 1.66 in Python 3.5.

**Fluorophore conjugation of ssLNA imaging probes.** In total 3′-amino ssLNA strands were purchased from Exiqon A/S and conjugated with ATTO 655-NHS (ATTO-TEC GmbH). Briefly, 10 nmol of ssLNA were mixed with ATTO 655-NHS in 1:5 molar ratio in 500 µL PBS, incubated for 2 h at room temperature and then overnight at 4 °C. Prior to purification with HPLC, isopropanol precipitation was performed to remove free dyes: 50 µL of 3 M sodium acetate solution was first added to the ssLNA solution followed by adding 550 µL of −20 °C isopropanol. The mixed solution was thoroughly vortexed and incubated for 30 min at −20 °C, and then immediately centrifuged at 10,000 × g and 4 °C for 1 h. The supernatant was delicately removed from the pellet. 500 µL of cold −20 °C ethanol was carefully added to the pellet and centrifuged at 10,000 × g and 4 °C for 15 min. The supernatant was removed, and the pellet was dried at 37 °C for 2 h. The pellet was then resuspended in 500 µL PBS, purified with HPLC, and lyophilized. See Supplementary Table 2 for ssLNA probe sequences.

**SDS-PAGE and mass spec validation of conjugated antibodies.** ssDNA-modified antibody solutions were reduced with 10 mM Tris-HCl complemented with 20 mM DTT for 2 h at 37 °C. Reduced antibody solutions were then run on a 10% acrylamide/bis-acrylamide SDS-PAGE gel, for 90 min at 110 V. Staining was performed using EZBlue gel staining reagent (Sigma-Aldrich). SiteClick conjugation efficiency and the ssDNA to antibody ratio were determined using MALDI-TOF mass spectrometry. Briefly, 20 µL of modified 0.5–1 mg/mL antibody solutions in PBS were purified and concentrated using ZipTip pipette tips C₄ resin (EMD Millipore) and then eluted in 10 µL of 80% ACN 0.1% TFA, dried down, and re-constituted in 1 µL of sinapinic acid matrix solution. The samples were then spotted and analyzed with microflex MALDI-TOF (Bruker Daltonics).

**Primary mouse and rat neuronal cultures.** Procedures for mouse neuronal culture preparation were approved by the Massachusetts Institute of Technology Committee on Animal Care. Hippocampal and cortical mouse neuronal cultures were prepared from postnatal day 0 or day 1 Swiss Webster (Taconic) mice as previously described[53,54] but with the following modifications: dissected hippocampal and cortical tissues were digested with 50 units of papain (Worthington) for 6–8 min, and the digestion was stopped with ovomucoid trypsin inhibitor (Worthington Biochem). Cells were plated at a density of 10,000 per well in a glass-bottom 96-well plate coated with 50 µl Matrigel (BD Biosciences). Neurons were

seeded in 50 µl plating medium containing MEM (Life Technologies), 33 mM glucose (Sigma-Aldrich), 0.01% transferrin (Sigma-Aldrich), 10 nM Hepes, 2 mM Glutagro (Corning), 0.13% Insulin (EMD Millipore), 2% B27 supplement (Thermo Fisher Scientific), and 7.5% heat inactivated fetal bovine serum (Corning). After cell adhesion, additional plating medium was added. 0.002 mM AraC (Sigma-Aldrich) was added when glia density was 50–70%. Neurons were grown at 37 °C and 5% $CO_2$ in a humidified atmosphere.

Procedures for rat neuronal culture were reviewed and approved for use by the Broad Institutional Animal Care and Use Committee. For rat hippocampal neuronal cultures, E18 embryos were collected from $CO_2$ euthanized pregnant Sprague Dawley rats (Taconic). Embryo hippocampi were dissected in ice-cold Hibernate E supplemented with 2% B27 supplements and 100 U/mL Penn/Strep (Thermo Fisher Scientific). Hippocampal tissues were digested in Hibernate E containing 20 U/mL papain, 1 mM L-cysteine, 0.5 mM EDTA (Worthington Biochem) and 0.01% DNAse (Sigma-Aldrich) for 8 min, and the digestion was stopped with 0.5% ovomucoid trypsin inhibitor (Worthington Biochem) and 0.5% bovine serum albumin (BSA) (Sigma-Aldrich). Neurons were dissociated and plated at a density of 15,000 cells/well onto poly-D-lysine coated, black-walled, thin-bottomed 96-well plates (Greiner Bio-One). Neurons were seeded and maintained in NbActiv1 (BrainBits). Cells were grown at 37 °C in a 95% air with 5% $CO_2$ humidified incubator for 21 days before use. For synaptic remodeling experiments, cells were treated on day 19 with 1 µM TTX (Sigma-Aldrich) for 48 h. All procedures involving animals were in accordance with the US National Institutes of Health Guide for the Care and Use of Laboratory Animals.

**Validation of ssDNA-conjugated antibodies and phalloidin**. To test whether the binding specificities of antibodies were affected by ssDNA conjugation, immunostaining patterns of unconjugated and ssDNA-conjugated antibodies were compared in each case. Cells were fixed at room temperature for 15 min with 4% paraformaldehyde (Electron Microscopy Sciences) and 4% wt/vol sucrose in PBS, and then washed three times with PBS. Cells were permeabilized for 10 min at room temperature with 0.25% Triton-X100 in PBS and washed twice with PBS. For staining with unconjugated primary antibody or fluorescently labeled phalloidin, cells were blocked for 1 h at room temperature with 5% BSA in PBS. Cells were then incubated with primary antibodies or fluorescently labeled phalloidin diluted in 5% BSA overnight at 4 °C. For staining with ssDNA-conjugated primary antibodies, the nuclear blocking buffer (5% BSA and 1 mg/mL salmon sperm DNA (Sigma-Aldrich) in PBS) was for blocking and antibody dilution. After primary antibody staining, the sample was then washed three times with PBS, incubated for 1 h at room temperature with secondary antibodies in 5% BSA, and washed again three times with PBS. For validation of ssDNA-conjugated secondary antibodies, the fluorescently labeled secondary antibodies of the same species were added to the samples after 30 min incubation with ssDNA-conjugated secondary antibodies to reduce the competition of binding of fluorescently labeled secondary antibodies with ssDNA-conjugated secondary antibodies. See Supplementary Tables 1 and 3 for antibody information. Comparison of antibody staining patterns before and after ssDNA conjugation were performed using the Pearson correlation coefficient (PCC). Colocalization of each antibody being tested with synapsin-I signal was performed before and after ssDNA conjugation. Specifically, three confocal images were acquired of neurons stained with unconjugated and conjugated antibodies separately. Each image was split into four quadrants, and the PCC between the synapsin-I channel and the channel of the other synaptic antibody for each quadrant was calculated and then averaged to obtain the mean PCC.

**Immunostaining for LNA- and DNA-PRISM**. Cells were fixed and permeabilized as described in the previous section. For LNA-PRISM, cells were additionally incubated in 50 µg/ml RNase A (Thermo Fisher Scientific) and 230 U/mL RNase T1 (Thermo Fisher Scientific) in PBS at 37 °C for 1 h to reduce the fluorescent background caused by ssLNA-RNA binding, and washed 3 times with PBS. Cells were then blocked for 1 h at room temperature with the 5% BSA in PBS. The following unconjugated primary antibodies were diluted in the regular blocking buffer and used for LNA- or DNA-PRISM: MAP2, VGLUT1, PSD-95, and NR2B (LNA-PRISM); PSD-95 and NR2B (DNA-PRISM). Cells were incubated in diluted primary antibodies overnight at 4 °C, washed 3 times with PBS, and then incubated in the nuclear blocking buffer for 1 h at room temperature. Next, the following secondary antibodies were diluted in the nuclear blocking buffer and used for LNA- or DNA-PRISM: goat-anti-chicken-Alexa 488, goat-anti-guinea pig-Alexa555 and goat-anti-rabbit-p1, goat-anti-mouse-p12 (LNA-PRISM); goat-anti-rabbit-p1 and goat-anti-mouse-p12 (DNA-PRISM). Cells were incubated at room temperature for 1 h in the secondary antibody solution. Cells were washed 3 times with PBS, post-fixed for 15 min with 4% paraformaldehyde and 4% wt/vol sucrose in PBS. This step was used to prevent cross-binding of the secondary antibodies to the primary antibodies in the following round of staining. Cells were washed 3 times with PBS and incubated again in the nuclear blocking buffer for 30 min at room temperature. Cells were then incubated overnight at 4 °C in the following primary antibody/peptide solution diluted in the nuclear blocking buffer for LNA- or DNA-PRISM: phalloidin-p2, Tuj-1-p3, cortactin-p4, SHANK3-p6, ARPC2-p7, bassoon-p8, synapsin-I-p9, Homer-1b/c-p10 (LNA-PRISM); phalloidin-p2, Tuj-1-p3, cortactin-p4, SHANK3-p6, bassoon-p5 (PNA), synapsin-I-p9, Homer-1b/c-p10 (DNA-PRISM). For the neuronal activity blockade experiment, the conjugated

antibody ARPC2-p7 was substituted for rat-Gephyrin with goat-anti-rat-p7 and the conjugated antibody Tuj-1-p3 was substituted for vGAT-p3. These inhibitory markers were excluded from analysis in the present work in order to present an in-depth study of both inhibitory and excitatory synapses using PRISM in future work. Cells were then washed 3 times with PBS. For LNA-PRISM, cells were incubated in diluted DAPI or Hoechst for 15 min. For DNA-PRISM, cells were incubated at room temperature for 1 h in donkey-anti-goat-Alexa488 diluted in the regular blocking buffer, washed 3 times with PBS, and then incubated with 10 nM of 100 nm diameter gold nanoparticle (Sigma-Aldrich) in PBS for 15 min. Cells were then washed 3 times with PBS prior to imaging. See Supplementary Table 1 and Supplementary Table 3 for antibody information.

**Multiplexed confocal imaging of neurons using LNA-PRISM**. LNA-PRISM imaging was performed on an Opera Phenix High-Content Screening System (PerkinElmer) equipped with a fast laser-based autofocus system, high NA water immersion objective (×63, numerical aperture = 1.15), two large format scientific complementary metal-oxide semiconductor (sCMOS) cameras and spinning disk optics. 405 nm, 488 nm and 561 nm lasers were used as excitation for DAPI, MAP2, and VGLUT1 channels respectively. PRISM images were acquired using a 640 nm laser (40 mW), sCMOS camera with 1–2 s exposure time, and effective pixel size of 187 nm. Before each imaging round, the corresponding imaging probe was freshly diluted to 10 nM in imaging buffer (500 mM NaCl in PBS, pH 8). Neurons were incubated with 10 nM imaging probes for 5 min, and then washed twice manually with imaging buffer to remove the free imaging probe. For each field of view, a stack of three images was acquired with a step of 0.5 µm. At least five fields of view were imaged for each well. After imaging, cells were washed three times with wash buffer (0.01 × PBS), and incubated in the wash buffer for 5 min after the last wash before the next round of imaging. To account for variabilities within each multi-well plate and across different cultures, imaging was performed on three independent neuronal cultures, and three wells were imaged from each neuronal culture. For the synaptic remodeling experiments, imaging was performed on six wells from two independent rodent neuronal cultures.

**LNA-PRISM confocal image processing and analysis**. The flat-field correction was performed to correct the uneven illumination in the images due to the laser beam profile[55]. Briefly, each image was filtered by morphological opening filter with a disk structural element of 100 pixels to estimate the background of each image. For each 96-well plate, all background images from the same channel were then averaged to obtain the illumination profile for each channel for each plate. Note that the illumination profile can vary across experiments; therefore, the illumination profile for each plate needs to be estimated separately. Images from the same channel were then divided by the illumination profile of the channel to obtain the corrected images. Next, lateral (x,y) drift between LNA-PRISM images from different imaging rounds was corrected by aligning the MAP2 channel in each imaging round. The (x,y) drift was estimated by locating the peak of the spatial cross-correlation function between two MAP2 images. For segmentation of synaptic punctae, the contrast of the image was first adjusted by saturating the highest and lowest 1% of pixels in the intensity histogram. The image was then denoised using a 5 × 5 Wiener filter, and filtered again with a top-hat operator with a disk structural element of 8 pixels to enhance the punctae in the image. The optimal threshold for each image was determined using an object-feature-based thresholding algorithm adapted from the thresholding algorithm previously used for single-molecule tracking[56]. The threshold producing the maximum number of objects was chosen as the optimal threshold. We found thresholding based on the features of objects was more robust to the intensity variations across different channels than the intensity-based approach for synapse segmentation. Connected synapses in the thresholded image were then separated using a watershed transform to obtain the final segmentation mask for each image of each synaptic target. Following Micheva et al.[12], synapses were identified using synapsin-I as the synapse marker. Synapsin-I punctae in the nuclei were mostly not synapses and therefore excluded from segmentation[12]. Each segmented synapsin-I punctum larger than 0.42 µm² was considered to be a synapse. For other synaptic proteins, only punctae that were colocalized (intensity weighted centroid distance <1 µm) with synapsin-I punctae were considered to be synapses and therefore retained. For each identified synapse, the integrated intensity and area of the segmented punctum in each synaptic channel were measured; zero was assigned when no colocalized punctum was detected. Average intensity was calculated by dividing the integrated intensity by the area. For non-synaptic targets (MAP2 and Tuj-1), the intensity was estimated by averaging the intensity within the synapsin-I punctae and no area measurement was performed. For the drug-treated experiments, the same measurements were. The punctae measurements were average for each image in a well. Values for each image in a well were averaged for each well ($n = 6$). Bar charts show the mean with points indicating independent values for each well. Error bars show 95% confidence intervals. P-values are calculated from a two-tailed Student's t-test. For the drug-free experiments, the Pearson correlation coefficient between each pair of synaptic intensity measurement was computed for each cell culture batch. The average correlation coefficients over three cell culture batches (total 178,528 synapses) were represented using a network diagram. An edge was shown between two nodes if the corresponding correlation was greater than 0.35, with the thickness of each edge representing the strength of the correlation. For the drug-

treated experiments the same method was used. Pearson correlation coefficient was calculated for each well and averaged to produce final results. For measuring differences between untreated and TTX treated groups, the difference from the mean untreated (for each biological replicate) was calculated for each well. The difference from the mean Pearson correlation coefficient was used to calculate confidence intervals and to perform two-tailed Student's $t$-test (Supplementary Data 2). t-Distributed Stochastic Neighbor Embedding (t-SNE) was used to visualize the distributions of synapses in high-dimensional feature space. 10,000 synapses were randomly subsampled from each replicate. 24 features (intensity levels from 13 channels and punctae sizes from 11 channels) of single synaptic profiles were used as the input to t-SNE. Each feature was log-transformed and normalized to have a standard deviation of one and minimum of zero before applying t-SNE. t-SNE analysis was performed using scikit-learn 0.18.1 in Python 3.5 with the exact method, perplexity parameter equal to 40, PCA initialization and 5000 iterations. The resulting t-SNE maps were similar for perplexity of 10–100. Hierarchical clustering of single-synapse profiles was performed by first normalizing the distribution of each feature to have a minimum of zero and a standard deviation of one using all synapses as input. For the synaptic remodeling experiments, 20,000 total synapses were randomly sub-sampled ($n = 10,000$ from untreated and $n = 10,000$ from TTX treated, $n = 2000$ from each replicate for each treatment group, $n = 5$ replicates) to produce the final t-SNE. 21 features (mean intensities and areas from ten synaptic targets and the mean intensity from MAP2 transformed as decribed above) were used as input. Hierarchical clustering was performed using 24 features (intensity levels from 13 channels and punctae areas from 11 channels) as input. Clustering was performed using the *clustermap* function from Seaborn 0.7.1 in Python 3.5 with the Euclidean metric and Ward's linkage.

**Multiplexed super-resolution imaging using DNA-PRISM.** Single and dual channel PAINT imaging was performed on an inverted Nikon Eclipse Ti microscope (Nikon Instruments) with the Perfect Focus System and oil-immersion objective (Plan Apo TIRF ×100, numerical aperture (NA) 1.49). A 642 nm wavelength laser (100 mW nominal) was used for excitation. Images were acquired using an Electron-Multiplying Charge-Coupled Device (EMCCD) camera (iXon DU-897, Andor Technology) with 100 ms exposure time, 100 EM gain, and effective pixel size of 160 nm. Nine-channel DNA-PAINT imaging was performed on an inverted Nikon Eclipse Ti microscope with the Perfect Focus System and oil-immersion objective. 640 nm laser (45 mW nominal) was used for excitation. Images were acquired using a Zyla 4.2 sCMOS camera (Andor Technology) with 100 ms exposure time, $2 \times 2$ binning, and effective pixel size of 130 nm. Cells were imaged using Highly Inclined and Laminated Optical illumination (HILO). The same imaging probe sequences labeled with Atto655 (Eurofins) and imaging/ washing buffer (500 mM NaCl in PBS, pH 8) were used as previously published[21] (Supplementary Table 2). Probe exchange was performed using a home-built fluid control system (Supplementary Note 3 and Supplementary Fig. 32). Depending on the labeling density, typically 0.5–3 nM imaging probe diluted in the imaging buffer was used in order to achieve optimal spot density for single-molecule imaging. 5000–20,000 image frames were typically acquired for each target.

**Super-resolution image reconstruction and analysis.** Localization of the center of each diffraction-limited spot corresponding to a single-molecule in the acquired movies was performed using DAOSTORM[57,58]. The super-resolution image was reconstructed by computing a 2D histogram of the (x,y) coordinates of the localized spots with bin size $5.4 \times 5.4$ nm, followed by smoothing with a 2D Gaussian filter. Non-specifically bound gold nanoparticles were used as fiducial markers to estimate the drift and align images of distinct targets. Drift was estimated by fitting splines to the x- and y-coordinates separately of each fiducial marker as a function of time using LOESS regression, and averaging over splines of all the fiduciary markers. 1D cross-sectional profiles of protein distributions were generated by projecting the 2D (x,y) coordinates of the localized spots onto the 1D coordinates along the trans-synaptic axes, and followed by computing a 1D histogram of the 1D coordinates. All image reconstruction and analysis procedures except for single-molecule localization were performed using MATLAB R2015a (MathWorks).

**Statistical analysis.** All statistical analysis was performed with R statistical environment[59]. Groups were compared using two-tailed Student's $t$-test using R[59]. $P$-values of <0.05 were considered significant. Graphs were generated using ggplot2[60] and ggpubr[61]. Detailed information on all statistical tests performed is listed in Supplementary Data 1 and Supplementary Data 2.

**Reporting summary.** Further information on research design is available in the Nature Research Reporting Summary linked to this article.

## Data availability
Representative image data generated during and/or analyzed during the current study are shown in Figures and Supplementary Materials. The datasets generated during and analyzed during the current study are available from the corresponding author on reasonable request.

## Code availability
The MATLAB and Python scripts used for image processing and analysis are available as Supplementary Software 1 and online at: https://github.com/lcbb/PRISM_analysis

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

## Acknowledgements

Funding from the NIH BRAIN Initiative Award 1U01MH106011 to M.B. and E.S.B., NIH R01-MH112694 to M.B. and J.R.C., and the NSF Physics of Living Systems 1305537 and 1707999 to M.B. is gratefully acknowledged. E.S.B. additionally acknowledges the HHMI-Simons Faculty Scholars Program, the Open Philanthropy Project, U. S. Army Research Laboratory and the U. S. Army Research Office under contract/grant number W911NF1510548, the New York Stem Cell Foundation-Robertson Award, and NIH grants 1RM1HG008525, 1R24MH106075, 1R01NS087950, and R01-MH112694. P.C.B. was supported by a Career Award at the Scientific Interface from the Burroughs Wellcome Fund. J.R.C. acknowledges funding from the Stanley Center for Psychiatric Research. Dr. Ralf Jungmann is acknowledged for helpful discussions on DNA-PAINT at the outset of this work. We acknowledge the Koch Institute Swanson Biotechnology Center, specifically Microscopy and Biopolymers & Proteomics core facilities, as well as the Whitehead Institute W.M. Keck Microscopy Facility, for technical support. Support for this research was provided by a core center grant (P30-ES002109) from the National Institute of Environmental Health Sciences, National Institutes of Health.

## Author contributions

S.-M.G., S.G., R.V., E.S.B., and M.B. conceived of and designed the DNA-PRISM study. S.-M.G. and M.B. conceived of the LNA-PRISM study. S.-M.G., L.L., J.R.C., and M.B. designed the LNA-PRISM study. R.V. designed and R.V. and E.W. and A.W. implemented the antibody conjugation protocols for DNA- and LNA-PRISM. S-M.G and S.G. designed and implemented the DNA-PRISM experiments. S.-M.G analyzed the DNA-PRISM data. D.P., K.P., and L.L. prepared primary neuronal cultures. S.-M.G., S.G., and R.V. designed and implemented the DNA-PRISM fixation, blocking, staining, probe exchange, and imaging protocols. S.-M.G. and L.L. adapted the DNA-PRISM protocol for LNA-PRISM. S.-M.G., L.L., E.D., K.P. and S.G. analyzed the LNA-PRISM data. S.G., A.B. K., and P.C.B. designed the microfluidic device for DNA-PRISM. M.B. designed and supervised the overall project including DNA-PRISM and LNA-PRISM. J.R.C. supervised the LNA-PRISM project. S.-M.G., S.G., and M.B. interpreted the DNA-PRISM results. S.-M.G., S.G., L.L., J.R.C., E.D., K.P., and M.B. interpreted the LNA-PRISM results. S.-M.G., S.G., E.D., and M.B. wrote the manuscript. All authors commented on the manuscript.

## Competing interests

P.C.B. is an extramural faculty member of MIT's Koch Institute for Integrative Cancer Research and a consultant to and equity holder in two companies in the microfluidics industry, 10X Genomics and General Automation Lab Technologies. The other authors declare no competing interests.
