## [Peer Review File · Nature Communications]

Editorial Note: This manuscript has been previously reviewed at another journal that is not operating a transparent peer review scheme. This document only contains reviewer comments and rebuttal letters for versions considered at *Nature Communications*. Mentions of the other journal have been redacted.

REVIEWERS' COMMENTS:

Reviewer #1 (Remarks to the Author):

Guo et al have carefully addressed comments from their previous round of review with **[REDACTED]** Although I am personally still not convinced by their super-resolution images of actin, overall the manuscript is suitable for Nature Communications, and I recommend publication.

However, I do notice that the authors have changed the title of the manuscript to "Quantitative neuronal phenotyping using multiplexed fluorescence imaging with diffusible probes". I don't feel this title is suitable –the results in here do not really address "quantitative neuronal phenotyping", but instead is mostly just about labeling and imaging. I thus urge the authors to go back to their previous title "Multiplexed, high-throughput neuronal fluorescence imaging with diffusible probes".

REVIEWERS' COMMENTS:

Reviewer #1 (Remarks to the Author):

Guo et al have carefully addressed comments from their previous round of review with **[REDACTED]** Although I am personally still not convinced by their super-resolution images of actin, overall the manuscript is suitable for Nature Communications, and I recommend publication. However, I do notice that the authors have changed the title of the manuscript to “Quantitative neuronal phenotyping using multiplexed fluorescence imaging with diffusible probes”. I don’t feel this title is suitable –the results in here do not really address “quantitative neuronal phenotyping”, but instead is mostly just about labeling and imaging. I thus urge the authors to go back to their previous title “Multiplexed, high-throughput neuronal fluorescence imaging with diffusible probes”.

We appreciate the positive assessment of our revision by the Reviewer, and we have changed the title back to its original form, in line with their request.